# Preventable Cancer in Israel

**DOI:** 10.3390/ijerph191710521

**Published:** 2022-08-24

**Authors:** Avi Magid, Eliezer Robinson, Lital Keinan-Boker

**Affiliations:** 1The Department of Information Systems, and the Department of Health Systems Management, The Max Stern Emek Yezreel College, Jezreel Valley 1930600, Israel; 2Faculty of Medicine, Technion—Israel Institute of Technology, Haifa 3200003, Israel; 3Israel Cancer Association, Givatayim 5310302, Israel; 4Israel Centre for Disease Control, Israel Ministry of Health, Ramat Gan 5262100, Israel; 5Faculty of Social Welfare and Health Sciences, School of Public Health, University of Haifa, Haifa 3498838, Israel

**Keywords:** cancer, Israel, modifiable risk factors, population-attributable-fraction, prevention

## Abstract

**Background:** The cancer burden in Israel is substantial. Recently calculated population attributable fractions (PAFs) for modifiable cancer risk factors in the US and the UK have indicated much room for prevention. Our aim was to estimate PAFs for selected modifiable cancer risk factors in Israel. **Methods****:** Exposure data on selected modifiable risk factors were based on national health surveys conducted in 2013–2015. Data on invasive cancer incidence in 2015 were retrieved from the Israel National Cancer Registry (INCR). Relative risks (RRs) for specific cancer sites were retrieved from the scientific literature. PAFs were calculated for selected modifiable risk factors using an approximate formula. **Results:** In 2015, 21% of all invasive cancer cases in Israel were attributed to smoking, 9% to excess body weight, and 1.5% to alcohol intake. Moreover, 32% of all invasive cancer cases were attributed to all modifiable risk factors (combined) included in our study. **Conclusions:** Quantifying the contribution of modifiable risk factors to the incidence of certain cancer types in Israel offers an opportunity for primary prevention and enables informed decision-making with respect to the prioritization of interventions.

## 1. Introduction

The cancer burden in Israel is substantial, with 29,586 newly diagnosed invasive cancer cases and 11,223 cancer deaths occurring in 2017 [1].

Cancer incidence in Jewish men has decreased by 3.1% per year between 2007–2017, whereas trends in Jewish women remained stable between 1990–2015; since 2015, cancer incidence in Jewish women has decreased by 6.7% per year. Cancer incidence in Arab men has decreased by 1.7% per year between 2006–2017, whereas in Arab women incidence rates slightly increased, at a rate of 0.8% per year between 1996–2017 [1].

Many cancer types are causally associated with modifiable risk factors such as smoking, excess body weight, alcohol consumption, red meat consumption, and others [2]. Recently, two studies estimated Population Attributable Fraction (PAFs) in the United States [2] and in the United Kingdom [3]. In the United States, it was estimated that 42% of all incident cancers in adults aged 30 years and older in 2014 were attributable to potentially modifiable cancer risk factors, where cigarette smoking had the highest PAF, followed by excess body weight, followed by alcohol intake, which was the third largest contributor to all cancer cases among women, and the fourth largest contributor among men. In addition, red and processed meat consumption accounted for 5.4% and 8.2% of colorectal cancer, respectively [2]. In the United Kingdom, it was estimated that 37% of all incident cancers in 2015 were attributable to potentially modifiable cancer risk factors, where, similar to the United States, cigarette smoking contributed the largest proportion of cancer cases, followed by overweight and obesity, which was the second largest preventable cause of cancer in the UK [3].

Estimating the PAFs of certain modifiable cancer risk factors for a specific population may be used as an effective tool for cancer control and prevention plans. 

In this paper we aimed to estimate PAFs for the ten most common cancer types in Israel in adults aged 30 years and older in 2015, with respect to potentially modifiable risk factors, namely cigarette smoking, excess body weight, alcohol intake, and red/processed meat consumption. 

## 2. Materials and Methods

### 2.1. Data Sources

#### 2.1.1. Modifiable Risk Factors 

Potentially modifiable cancer risk factors were identified using the literature [2,3]. We focused on factors for which exposure data in Israel were available and comparable. These risk factors included active cigarette smoking, excess body weight, alcohol intake, red meat consumption, and processed meat consumption. 

Cigarette smoking was defined based on self-report as a variable with three categories: current, former, never. Excess body weight was defined based on measurements as body mass index (BMI, kg/m^2^) ≥ 25.0. Alcohol intake was defined based on self-report as a variable with three categories: ≤12.5 g/d, 12.51–50 g/d, and >50 g/d. Red meat consumption was defined as a variable with six categories: <100 g/d, 100–<200 g/d, 200–<300 g/d, 300–<400 g/d, 400–<500 g/d, 500 g/d and above. Processed meat consumption was defined as a variable with six categories: <100 g/d, 100–<200 g/d, 200–<300 g/d, 300–<400 g/d, 400–<500 g/d, 500 g/d and above.

We identified the ten common cancer types in Israel. These types include breast cancer (women), colorectal cancer, lung cancer, bladder cancer, thyroid cancer, corpus uteri cancer, kidney cancer, pancreatic cancer, stomach cancer, and ovary cancer [1]. Table 1 summarizes factors associated with increased cancer risk by cancer type (ICD-10).

#### 2.1.2. Prevalence of Exposure

Nutritional exposures were retrieved from the MABAT survey of 2014–2016, the latest in a series of national cross-sectional studies on the status of health and nutrition of the Israeli population, carried out on a representative sample of the population by age groups. A total of 5113 subjects aged 18–64 were invited to participate in the 2014–2016 MABAT survey; 2957 of them consented (response rate 57.8%). The participants filled out a general questionnaire relating to health behaviours and chronic morbidity, as well as a 24 h dietary recall (with 20% of the participants also filling out a food frequency questionnaire). The survey aimed to estimate the daily average nutritional consumption of macro- and micronutrients, as well as food items such as alcohol, red meat, and processed meat [4]. It was approved by the Supreme Institutional Review Board of the Ministry of Health (MOH-043-2012) and all participants signed informed consent forms.

Exposure data on other factors were retrieved from the Israel National Health Interview Survey 3 (INHIS–3), which was carried out in 2013 through 2015 on a representative sample of the Israeli population aged 21 and over. The INHIS-3 cross-sectional study included 4511 residents of Israel aged 21 years and older (response rate 24.6%). The study aimed to estimate the prevalence of chronic diseases and conditions, to obtain information on the utilization of health services, and to estimate the prevalence of health behaviour patterns such as smoking and excess body weight [5]. The INHIS studies are carried out periodically by the Israel Centre for Disease Control as part of the regulatory activity of the Ministry of Health aimed to assess population health status and policy implementation. Thus, they do not require an IRB clearance. Participants are not identified.

#### 2.1.3. Cancer Incidence

Data on new invasive cancer cases in Israel were retrieved from the Israel National Cancer Registry (INCR) for the year 2015, which was the year for which data were completed at the INCR at the time of the study. The INCR was established in 1960, and since 1982, reporting to the INCR on all newly diagnosed cancer cases is mandatory. Data collected by the INCR include demographic information (sex, date of birth, country of birth, date of immigration to Israel if applicable), date and location of cancer diagnosis, histological type and grade of malignant tumour, and disease stage at diagnosis. Completeness of this registry is estimated at approximately 96.5% for solid tumours [6].

#### 2.1.4. Relative Risks

Relative risks (RRs) associated with the risk factors defined for specific cancer types were retrieved from the scientific literature, mostly based on the two recently published studies which estimated PAFs in the United States [2] and in the United Kingdom [3].

### 2.2. PAFs Calculation

Israel PAFs in adults aged 30 years and older were estimated using the following approximate formula:$$PAF=\frac{\sum P_{i}\left(RR_{i}-1\right)}{\sum P_{i}\left(RR_{i}-1\right)+1}$$
where $P_{i}$ is the exposure prevalence at the exposure category i, and $RR_{i}$ is the corresponding RR [2]. The number of cancer cases attributable to each risk factor was achieved by multiplying the PAF for a specific cancer type and risk factor by the number of cases of this cancer type occurring in 2015.

Whenever there were several risk factors associated with a specific cancer type, we assumed no interaction when calculating, by summing up, the overall attributable proportion and number of cases for this cancer type.

We estimate the PAF for the entire population as well as separately for Jews and Arabs. The reason for this approach is the differences between exposure prevalence and outcome incidence between these two population groups. We compared our results to previously published results. 

In addition, we calculated the percentage of all cancer cases in Israel in 2015 attributed to each of the exposures included in our study.

All analyses were done using MATLAB software.

No ethical approval was needed since only previously published, aggregated data were used.

## 3. Results

Our results are presented in Table 2, Table 3 and Table 4.

Approximately 32% of all common cancer cases in 2015 (5473 cases) were attributed to modifiable risk factors (combined) included in our study: cigarette smoking, excess body weight, alcohol intake, red meat consumption, and processed meat consumption.

### 3.1. Cigarette Smoking

According to the INHIS-3 survey for 2013–2015, 18% of the sample was current smokers, and 25% of the sample reported current and former smoking. The prevalence of current smoking among Israeli men was 22%, and among Israeli women was 15%. The prevalence of former smoking among Israeli men was 34%, and among Israeli women was 16%. Comparison between Jews and Arabs in Israel shows dramatic differences in smoking prevalence: among Arab men, smoking prevalence was 2.8 times higher than among Jewish men; Jewish women’s smoking prevalence was twofold higher than that of Arab women.

Of all modifiable risk factors evaluated in our study, cigarette smoking accounted for the highest proportion and the largest number of attributed cancer cases (21% of all common cancer cases in 2015). Lung cancer in particular had the highest proportion attributable to smoking (85%), followed by bladder cancer (46%), stomach cancer (18%), kidney cancer (16%), pancreatic cancer (13%), and colorectal cancer (12%) (Table 2).

### 3.2. Excess Body Weight

According to the INHIS-3 survey for 2013–2015, the prevalence of excess body weight (BMI of 25 kg/m^2^ and over) was 61%, where 20% of the sample population had BMI ≥ 30 kg/m^2^. Comparison between Jews and Arabs in Israel shows that excess body weight prevalence among Arab women was 1.39 times higher than excess body weight prevalence among Jewish women.

Excess body weight was associated with 9% (1566 cases) of all common cancer cases diagnosed in 2015 in Israel. Corpus uteri cancer, in particular, had the highest proportion attributable to excess body weight (30%, 228 cases), followed by kidney cancer (26%), stomach cancer (18%), colorectal cancer (13%), pancreatic cancer (9%), thyroid cancer (9%), female breast cancer (8%), and ovary cancer (5%) (Table 2).

### 3.3. Alcohol Intake

According to the MABAT survey for 2014–2016, 95% of the sample population consumed less than 12.5 grams of ethanol per day.

Alcohol intake was associated with 1.5% (259 cases) of all common cancer cases diagnosed in 2015 in Israel. Only two cancer types were attributed to alcohol intake: colorectal cancer (4%, 229 cases), and female breast cancer (1%, 30 cases) (Table 2).

### 3.4. Red Meat Consumption

According to the MABAT survey for 2014–2016, 85% of the sample population consumed less than 100 grams of red meat per day.

Red meat consumption was associated with only 2% (54 cases) of colorectal cancer diagnosed in 2015 in Israel (Table 2).

### 3.5. Processed Meat Consumption

According to the MABAT survey for 2014–2016, 85% of the sample population consumed less than 100 grams of processed meat per day.

Processed meat consumption was associated with only 0.4% (12 cases) of colorectal cancer, and 0.5% (four cases) of stomach cancer diagnosed in 2015 in Israel (Table 2).

### 3.6. Comparison between Jews and Arabs in Israel

Our results show greater fractions attributable to cigarette smoking and excess body weight for certain cancer types in the Arab population compared to the Jewish population in Israel. For example, 16% of pancreas cancer cases in the Arab population in 2015 were attributed to smoking compared to 13% in the Jewish population; 38% of corpus uteri cancer cases in the Arab population in 2015 were attributed to excess body weight compared to 28% in the Jewish population; 11% of breast cancer cases in the Arab population in 2015 were attributed to excess body weight compared to 7% in the Jewish population (Table 3 and Table 4).

## 4. Discussion

In this study we aimed to assess the population attributable fraction of selected modifiable risk factors with respect to invasive cancer incidence. Our results showed that for many of the common cancer cases in Israel in 2015, above 25% of the cases were attributable to modifiable and potentially preventable risk factors (Table 5). Moreover, our results demonstrated that an estimated one third (32%) of all common cancer cases in Israel in 2015 were attributable to modifiable and potentially preventable risk factors (Table 6). In other words, these cases could have been prevented by behavioural change. Our results showed that a higher percentage of cancer cases are attributable to smoking and excess body weight in the Arab population compared to the Jewish population. This difference may be partially due to the higher frequency of smoking among Arab males (41%), and higher frequency of excess body weight in Arab males (72%) and females (71%), compared to Jewish subjects: 18%, 67% and 51%, respectively.

According to our results, cigarette smoking and excess body weight were responsible for the highest percentage of preventable cancer cases in Israel in 2015, and therefore current efforts for primary prevention interventions should be continued and fortified.

### 4.1. Cigarette Smoking

Despite the decreasing trend in smoking prevalence in Israel [7], cigarette smoking has been found to be the leading contributor to cancer cases in Israel in 2015, accounting for 20.9% of all cancer cases. It is noteworthy that we did not estimate the contribution of second-hand smoking to cancer cases in Israel due to insufficient and mostly non-quantitative exposure data. Therefore, the burden of cancer attributable to smoking in Israel may be even higher than our estimates. Our results point out that applying comprehensive tobacco-control programs may significantly reduce the overall cancer burden in Israel, particularly with respect to lung, pancreas, stomach, kidney, colorectal, and bladder cancers (Table 2). Among the proven methods for reducing smoking prevalence are taxation on tobacco products, smoke-free laws (legislation and enforcement), and assistance with smoking cessation [2]. According to the Israeli Minister of Health report on smoking for 2018, some tobacco-control programs have been applied in Israel. Taxation on tobacco products was proven to have the strongest effect on smoking cessation in the US, with a higher impact on lower income people, who have higher smoking rates, and on teenagers, who may not start smoking due to taxation [7]. In recent years, a taxation policy on cigarettes and other tobacco products has been applied. In addition, following a ruling of the Israeli Supreme Court, the taxation policy was also applied to tobacco sold for self-made cigarettes. However, enforcement should be improved, in particular, enforcement of selling single (not packaged) cigarettes (which are sold especially to the youth), and tobacco for self-made cigarettes [7]. Legislation of smoke-free laws in public areas was updated and has been recently expanded to also apply in children’s playgrounds, zoos, sport facilities, in adjacency to kindergartens, educational institutes (including buildings, yards, and within ten metres of the school entrance), and more. However, enforcement of these smoke-free laws is still low [7]. An emphasis should be put on enforcing these laws, thereby reducing the damage caused by smoking. In addition, the Israeli Ministry of Health started a campaign for reducing smoking and encouraging smoking cessation in the Arab population in 2018, including in the Arab press and social media [7]. Such campaigns, as well as tailored intervention programs for smoking cessation in the Arab population, should be continued to reduce the gap in smoking prevalence between the Arab population and the Jewish population in Israel, and consequently also to reduce the burden of cancer attributable to smoking in the Arab population. Importantly, medications and workshops for smoking cessation have been already included for a few years in basic health care services in Israel, and they are accessible to all [7]. However, tailored intervention programs are necessary to promote their usage.

### 4.2. Excess Body Weight

We estimated that 9% of all cancer cases in Israel were attributable to excess body weight. These findings emphasise the importance of adherence to weight control guidelines and recommendations. Overweight and obesity are considered an epidemic in Israel. Currently, nearly 60% of Israeli adults and 7% of Israeli children aged 5–18 years are overweight or obese [8]. Children overweight and obesity may be extended into adulthood and increase the risk for obesity-related diseases [2]. Therefore, national programs on weight control should strongly focus on children with excess body weight. Such programs may be school-based as well as community-based interventions targeted at promoting weight control, healthy diet, and physical activity. Weight control promotion interventions for adults with excess body weight should also be recommended to decision makers.

### 4.3. Alcohol Intake, Red Meat Consumption, and Processed Meat Consumption

Although alcohol intake, red meat consumption, and processed meat consumption (combined) were estimated to attribute only nearly 2% of all cancer cases in Israel, it is important to increase the public awareness regarding the potential harm they may cause. Special emphasis should be put on the causal association with specific cancer types, especially the association between alcohol consumption and female breast cancer, and the association between red meat and processed meat consumption with colorectal cancer.

### 4.4. Comparison to Previously Published Results

Our results show that compared to the US, smoking in Israel is a stronger cancer-causing factor: 85% of lung cancer cases in Israel were attributed to smoking in 2015 compared to 82% in the US; 13% of pancreas cancer cases in Israel were attributed to smoking in 2015 compared to 10% in the US (Table 2).

On the other hand, in most of the related cancer types, excess body weight, alcohol consumption, red meat consumption, and processed meat consumption were found to be responsible for larger fractions of preventable cancer cases in the US compared to Israel. A dramatic example is that excess body weight was responsible for 60% of corpus uteri cancer cases in the US in 2015, compared to 30% cases in Israel in 2015, as demonstrated by our results.

The differences between our results and the US results with respect to cigarette smoking may be due to a lower smoking rate in the US compared to Israel in the last decades. Smoking prevalence for the entire US population in 2017 was 14% [9], whereas smoking prevalence for the entire Israel population in 2017 was 19.8%. Smoking prevalence in the US has dramatically decreased over the last decade, as a result of excise taxes on cigarettes, and expansion of the law which prohibits smoking in public areas [2] as well as greater enforcement. Smoking prevalence in Israel has also decreased [7] but later than in the US and to a lesser extent. In addition, enforcement of smoke-free laws in Israel is low. Moreover, smoking prevalence in the US has been persistently lower compared to smoking prevalence in Israel over the last 20 years [10]. This explains the lower percentage of cases attributed to smoking in the US compared to Israel; cancer is a group of diseases mostly characterized by a long latency period, and as such, the differences observed in 2015 reflect differences in exposures twenty or more years before. 

Comparing our results to the UK results, one can observe that smoking in Israel was responsible for 20.9% of cancer cases in 2015 (Table 6), compared to 17.7% of cancer cases in the UK [3]. This difference may also be due to a lower smoking rate in the UK compared to Israel in the last decade. Smoking prevalence for the entire UK population in 2017 was 15.1% [10], compared to 19.8% in Israel.

The differences between our results and the US results with respect to excess body weight, alcohol consumption, red meat consumption, and processed meat consumption may be due to higher rates of overweight and obesity in the US compared to Israel (currently in the US nearly 75% of adults are overweight or obese, whereas in Israel nearly 60% of adults are overweight or obese) [10,11], and due to higher consumption rates of alcohol, red meat, and processed meat in the US compared to Israel [12,13]. Red meat and processed meat are risk factors with strong evidence of causing colorectal cancer in humans. Moreover, according to the International Agency for Research on Cancer (IARC), red meat and processed meat are classified as type 2A and type 1 carcinogens, accordingly [14]. However, our results showed that these factors were responsible for a very low fraction of cases in 2015 in Israel. These findings may be explained by the relatively low consumption of red meat and processed meat in Israel [15].

Excess body weight in Israel was responsible for 9.1% of cancer cases in 2015 (Table 6), where in the UK excess body weight was responsible for only 6.3% of cancer cases [3]. This may also be due to lower excess body weight rates in the UK compared to Israel during the last decades [11].

We observed some differences between Israel and the UK with respect to alcohol consumption (responsible to 1.5% of cancer cases in Israel in 2015 vs. 3.3% in the UK in 2015), and processed meat consumption (responsible for 0.09% of cancer cases in Israel in 2015 vs. 1.5% in the UK in 2015). These differences may also be due to lower prevalence of these exposures in Israel during the years [12,13].

Our study is based on national data, and as such, the validity of the exposure and the outcome is high.

Our study also has some limitations. First, our study deals with only a few risk factors compared to the US and the UK studies. In particular, due to insufficient data, our study does not deal with the effects of sedentary lifestyle on various cancer types, and therefore we may underestimate the fraction of preventable cancer cases in Israel. Second, whenever we calculated the overall attributable proportion, and the number of preventable cancer cases for a given cancer type with several risk factors, we assumed that there was no interaction between these risk factors, and that exposure at the relevant time window was well reflected by the exposure at the time it was measured. Third, the estimates of the attributable risk come from non-randomised studies. This may affect the quality of the data, however, non-randomized studies are the only way to estimate the attributable risk, since randomized studies which involve exposures such as smoking, alcohol consumption, and nutrition are usually not performed. In order to increase the quality assurance of our study, we compared our results with previous studies, which estimated the attributable risk based on the same type of data. Another limitation is that the surveys used to estimate the exposures have a low response rate, in particular the estimate of alcohol consumption. This may lead to an under estimation of the PAFs. However, it is important to mention that alcohol consumption in Israel is low compared to other western countries. Therefore, even if the low response rate affects the PAF estimation, its effect is minor. Finally, our study does not take into account the long latency period of the various cancer types.

## 5. Conclusions

Based on this study, a third (32%) of the total burden of cancer in Israel is attributable to modifiable risk factors, and is potentially preventable by behavioural changes. Our findings point out that lifestyle and unhealthy behaviours such as smoking, excess body weight, alcohol intake, and red meat/processed meat consumption significantly affect cancer morbidity in Israel. Many cancer cases in Israel could have been prevented by effective preventive strategies. Therefore, our results emphasize the continuous need for prevention actions in order to reduce the morbidity from cancer associated with potentially modifiable risk factors. Our study points out that priority should be given to prevention programs on cigarette smoking cessation and weight control.

Further research is needed to assess PAFs for other modifiable risk factors such as exposure to second hand smoking, physical inactivity, low fruit/vegetables consumption, low dietary fibre consumption, and low dietary calcium consumption. In addition, such analyses should be repeated periodically in order to assess the effect of targeted intervention on cancer burden.

To summarize, quantifying the contribution of modifiable risk factors to the incidence of certain cancer types offers an opportunity for primary prevention of cancer and enables informed decision-making with respect to prioritization of interventions aimed to modify the exposure to these factors. Such prioritization of interventions may also be tailored to specific population groups in Israel (Jews and Arabs).

## Figures and Tables

**Table 1 ijerph-19-10521-t001:** Factors associated with increased cancer risk (by cancer type, ICD-10) used in this study.

Risk Factor	Cancer Type (ICD-10)
Smoking	Stomach (C16); colorectum (C18-C20, C26.0); pancreas (C25); lung (C33-C34); kidney (C64-C66); urinary bladder (C67)
Excess body weight	Stomach (C16); colorectum (C18-C20, C26.0); pancreas (C25); female breast (C50); corpus uteri (C54-C55); ovary (C56); kidney (C64-C66); thyroid (C73)
Alcohol intake	Colorectum (C18-C20, C26.0); female breast (C50)
Red meat consumption	Colorectum (C18-C20, C26.0)
Processed meat consumption	Colorectum (C18-C20, C26.0); stomach (C16)

**Table 2 ijerph-19-10521-t002:** Estimates cancer cases in adults aged 30 years and older in Israel in 2015 attributable to potentially modifiable risk factors by sex, risk factor, and cancer type, compared to the US.

	MEN		WOMEN		BOTH	
Cancer	% PAF (95% CI)	Attributable Cases	% PAF (95% CI)	Attributable Cases	% PAF (95% CI)	Attributable Cases
**Cigarette Smoking**						
Lung	87 (83.5–90.5)	1375	81 (76.8–85.2)	738	85 (81.7–88.3)	2104
Pancreas	16 (12.1–19.9)	72	11 (8.2–13.8)	46	13 (11.3–14.7)	115
Stomach	22 (19.8–24.2)	94	14 (10.3–17.7)	40	18 (15.6–20.4)	128
Kidney	20 (15.1–24.9)	115	13 (10.9–15.1)	35	16 (12.7–19.3)	139
Colorectum	15 (13.9–16.1)	233	9 (8.1–9.9)	149	12 (11.1–12.9)	378
Bladder	52 (45.9–58.1)	652	40 (35.5–44.5)	117	46 (42.2–49.8)	714
**Excess body weight**						
Corpus uteri	--	--	30 (27.2–32.8)	228	30 (27.2–32.8)	228
Kidney	18 (16.4–19.6)	109	24 (23.2–24.8)	67	26 (25.5–26.5)	219
Stomach	19 (17.7–20.3)	80	16 (13.9–18.1)	46	18 (16.3–19.7)	124
Pancreas	10 (9.6–10.4)	46	7 (6.3–7.7)	27	9 (8.1–9.9)	80
Thyroid	9 (7.6–10.4)	23	8 (7.1–8.9)	62	9 (7.7–10.3)	87
Breast	--	--	8 (6.9–9.1)	423	8 (6.9–9.1)	423
Colorectum	14 (13.2–14.8)	207	5 (4.9–5.1)	83	13 (11.6–14.4)	387
Ovary	--	--	5 (4.7–5.3)	18	5 (4.7–5.3)	18
**Alcohol intake**						
Breast	--	--	4 (3.6–4.4)	229	4 (3.6–4.4)	229
Colorectum	1 (0.8–1.2)	23	0.3 (0.1–0.5)	5	1 (0.8–1.2)	30
**Red meat consumption**						
Colorectum	2 (1.5–2.5)	34	1 (0.8–1.2)	22	2 (1.7–2.3)	54
**Processed meat consumption**						
Colorectum	0.6 (0.2–1.0)	8	0.3 (0.1–0.5)	4	0.4 (0.1–0.7)	12
Stomach	0.7 (0.2–1.2)	3	0.4 (0.1–0.7)	1	0.5 (0.3–0.7)	4

**Table 3 ijerph-19-10521-t003:** Estimates cancer cases in Jewish adults aged 30 years and older in Israel in 2015 attributable to potentially modifiable risk factors by sex, risk factor, and cancer type.

	MEN		WOMEN		BOTH	
Cancer	% PAF (95% CI)	Attributable Cases	% PAF (95% CI)	Attributable Cases	% PAF (95% CI)	Attributable Cases
**Cigarette Smoking**						
Lung	86 (82.3–89.7)	1068	82 (78.5–85.5)	680	85 (81.2–88.8)	1747
Pancreas	14 (12.4–15.6)	56	12 (10.8–13.2)	43	13 (11.6–14.4)	100
Stomach	20 (19.8–20.2)	74	16 (14.3–17.7)	36	18 (16.4–19.6)	108
Kidney	19 (16.6–21.4)	94	14 (11.9–16.1)	31	17 (14.8–19.2)	120
Colorectum	14 (13.2–14.8)	182	11 (10.4–11.6)	138	13 (12.3–13.7)	323
Bladder	51 (46.6–55.4)	540	42 (38.1–45.9)	110	47 (43.0–51.0)	621
**Excess body weight**						
Corpus uteri	--	--	28 (26.9–29.1)	173	28 (26.9–29.1)	173
Kidney	18 (17.1–18.9)	90	23 (21.9–24.1)	53	25 (23.7–26.3)	181
Stomach	18 (15.6–20.4)	66	15 (13.1–16.9)	36	17 (15.0–19.0)	102
Pancreas	10 (9.3–10.7)	40	6 (5.7–6.3)	23	7 (6.6–7.4)	58
Thyroid	9 (7.6–10.4)	20	8 (6.4–9.6)	47	8 (6.4–9.6)	71
Breast	--	--	7 (6.7–7.3)	344	7 (6.7–7.3)	344
Colorectum	13 (11.1–14.9)	171	5 (4.6–5.5)	67	12 (10.8–13.2)	322
Ovary	--	--	4 (3.8–4.2)	14	4 (3.8–4.2)	14
**Alcohol intake**						
Breast	--	--	4 (3.6–4.4)	198	4 (3.6–4.4)	198
Colorectum	1 (0.97–1.03)	19	0.3 (0.28–0.32)	4	0.1 (0.07–0.13)	26
**Red meat consumption**						
Colorectum	2 (1.95–2.05)	26	1 (0.97–1.03)	17	2 (1.95–2.05)	43
**Processed meat consumption**						
Colorectum	0.7 (0.66–0.74)	7	0.3 (0.27–0.33)	3	0.4 (0.35–0.45)	10
Stomach	0.7 (0.68–0.72)	3	0.4 (0.35–0.45)	1	0.5 (0.49–0.51)	4

**Table 4 ijerph-19-10521-t004:** Estimates cancer cases in Arab adults aged 30 years and older in Israel in 2015 attributable to potentially modifiable risk factors by sex, risk factor, and cancer type.

	MEN		WOMEN		BOTH	
Cancer	% PAF (95% CI)	Attributable Cases	% PAF (95% CI)	Attributable Cases	% PAF (95% CI)	Attributable Cases
**Cigarette Smoking**						
Lung	91 (86.5–95.5)	217	67 (63.5–70.5)	32	86 (81.8–90.2)	246
Pancreas	24 (21.1–26.9)	7	6 (4.9–7.1)	2	16 (14.7–17.3)	9
Stomach	28 (25.5–30.6)	10	7 (5.2–8.8)	2	19 (15.6–22.4)	11
Kidney	24 (21.2–26.8)	13	6 (5.1–6.9)	1	15 (11.9–18.1)	12
Colorectum	20 (19.1–20.9)	32	5 (4.8–5.2)	7	13 (12.3–13.7)	39
Blader	60 (56.7–63.3)	71	22 (19.1–24.9	4	47 (44.0–50.0)	64
**Excess body weight**						
Corpus uteri	--	--	38 (35.8–40.2)	33	38 (35.8–40.2)	33
Kidney	20 (18.1–21.9)	11	31 (29.3–32.7)	8	30 (28.8–31.2)	23
Stomach	20 (17.2–22.8)	7	22 (20.9–23.1)	6	21 (19.5–22.5)	13
Pancreas	11 (10.4–11.6)	3	9 (8.1–9.9)	2	11 (10.4–11.6)	6
Thyroid	10 (9.3–10.7)	2	11 (10.4–11.6)	12	11 (10.4–11.6)	13
Breast	--	--	10 (9.6–10.4)	51	10 (9.6–10.4)	51
Colorectum	15 (12.4–17.6)	23	8 (6.3–9.7)	11	15 (12.4–17.6)	46
Ovary	--	--	6 (5.1–6.9)	2	6 (5.1–6.9)	2
**Alcohol intake**						
Breast	--	--	4 (3.6–4.4)	20	4 (3.6–4.4)	20
Colorectum	1 (0.97–1.03)	2	0.1 (0.08–0.12)	0	0.8 (0.77–0.83)	2
**Red meat consumption**						
Colorectum	3 (2.9–3.1)	4	2 (1.8–2.2)	3	2 (1.8–2.2)	7
**Processed meat consumption**						
Colorectum	0.5 (0.47–0.53)	1	0.4 (0.38–0.42)	1	0.4 (0.38–0.42)	2
Stomach	0.7 (0.67–0.73)	0	0.5 (0.48–0.52)	0	0.5 (0.48–0.52)	0

**Table 5 ijerph-19-10521-t005:** Estimated proportion and number of incident cancer cases attributable to all evaluated risk factors and estimated total number of cancer cases in adults aged 30 years and older in Israel in 2015, by cancer type.

Cancer	% PAF (95% CI)	Attributable Cases	Total No. of Cases in 2015
Lung	84.8 (81.5–88.1)	2104	2482
Pancreas	22.5 (20.3–24.7)	195	867
Stomach	36.2 (33.8–38.6)	256	708
Kidney	42.0 (37.9–46.1)	358	853
Colorectum	28.3 (27.0–29.6)	861	3047
Bladder	46.4 (44.3–48.5)	714	1539
Corpus uteri	29.7 (27.8–31.6)	228	767
Thyroid	8.6 (7.8–9.4)	87	1008
Breast	12.0 (11.3–12.7)	652	5459
Ovary	4.6 (4.0–5.2)	18	392

**Table 6 ijerph-19-10521-t006:** The proportion and number of incident cancer cases (out of all cancer) in adults aged 30 years and older in Israel attributable to each evaluated risk factor, by risk factor.

Risk Factor	% PAF (95% CI)	Attributable Cases
Smoking	20.9 (19.7–22.1)	3579
Excess body weight	9.1 (8.2–10.0)	1566
Alcohol intake	1.5 (1.1–1.9)	259
Red meat consumption	0.3 (0.26–0.34)	54
Processed meat consumption	0.09 (0.04–0.14)	16
All factors combined	32.0 (28.8–35.2)	5473

## Data Availability

The dataset generated and/or analysed during the current study is available from the corresponding author on reasonable request.

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
