# Peer review of "Preventable Cancer in Israel"

_ijerph, 2022, doi:10.3390/ijerph191710521_

Round 1
Reviewer 1 Report
Thank you for inviting me to review this study "Preventable Cancer in Israel". In the paper, the authors use survey and surveillance data to estimate the prevalence of cancer and cancer risk factors, and effect estimates from the existing literature, to estimate the population attributable cancer risk for various exposures. This answers an important question for the Israeli public health service; and the analysis looks competently executed and implemented.
My one comment would be that a larger discussion of the quality of data used to estimate the attributable risk would be good: Assuming the effect estimates come from non-randomised studies, what potential effect could residual biases have (are they concordant with the effect estimates from quasi-experimental methods like Mendelian randomisation, for example?), likewise, the survey had a low response rate. Could this lead to error (e.g. if excess alcohol consumption results in a reduced probability of response you will be under estimating the PAF).
Author Response
Thanks a lot for reviewing my article.
Indeed, the estimate of the attributable risk comes for non-randomized studies. However, this is the only way to estimate the attributable risk, since randomized studies which involve exposures such as smoking, alcohol consumption and nutrition are usually not performed. Indeed, this is an important limitation which I added to the discussion section. In order to increase the quality assurance of our study, we compared our results with previous studies, which estimated the attributable risk based on the same estimates.
As for the low response rate of the survey (in particular, the estimation of alcohol consumption), The response rate is indeed low, but this low rate is acceptable in such surveys. I added it as another limitation in the discussion section of this paper. It is important to mention that alcohol consumption in Israel compared to other western countries is low, and therefore, even if the low rate affects the result, its effect is minor.
Reviewer 2 Report
Congratulations on the research idea
We recomend continuing assessments how these risk estimaties have influenced the prevalence of modifiable factors as a result of national prevention strategies
The topic of the manuscript is of the major interest and is a very challenging issue in the field. However, there are some minor issues that have to be addressed:
I. Page 1, “Abstract”, rows 17, 18, 19:
- “PAFs were calculated for selected modifiable risk factors using the following approximate formula: ???= Σ??(???−1)/Σ??(???−1)+1, where ?? is the exposure prevalence at the exposure category ?, and ??? is the corresponding RR.”
· Comments: Maybe the authors want to consider rephrasing the paragraph because it is not recommended to use mathematical formulas in the Abstract.
II. Pages 2-3, “Materials and Methods” Section:
- The authors provided definitions for all the variables involved in the research: modifiable risk factors for cancers and types of cancer.
- For the statistical analysis there are some information well presented (prevalence of the exposure, incidence of cancers, relative risk or population attributable fractions for modifiable cancer risk factors).
- Row 57: The authors used: "Potentially modifiable cancer risk factors were identified using the literature (2) (3)."
INSTEAD OF
- "Potentially modifiable cancer risk factors were identified using the literature (2, 3)."
· Comments: However, the authors should take into consideration to provide the statistical significance (p value) and the confidence interval (CI) for their results.
III. Page 7, “Discussion” Section:
- The authors have identified and addressed the limitations of the study.
· Comments: From the point of view of the research methodology there is not allowed to use tables and figures in this section. There have to be used in the "Results" Section.
IV. Pages 10-11, “References” Section.
- The references are appropriate as relevance, number and year of publication.
· Comments:
- Some references are not available in English language (no. 1, 4, 5, 7).
- Some references have to be appropriate written (8, 11, 12, 13, 14).
- Maybe the authors wish to take into consideration to revise the References Section as indicated by the journal rules:
For example:
2. Islami, F., Goding, S.A., Miller, K.D., Siegel, R.L., Fedewa, S.A., Jacobs, E.J. et al. Proportion and number of cancer cases and 366 deaths attributable to potentially modifiable risk factors in the United States. CA Cancer J. Clin. 2018; 68(1), 31-54. doi:10.3322/caac.21440.
3. Brown, K.F., Rumgay, H., Dunlop, C., Ryan, M., Quartly, F., Cox, A. et al. The fraction of cancer attributable to modifiable risk 369 factors in England, Wales, Scotland, Northern Ireland, and the United Kingdom in 2015. Br. J. Cancer 2018; 118(8), 1130-1141. doi:10.1038/s41416-018-0029-6.
9. Centers for Disease Control and Prevention. Current cigarette smoking among adults in the United State. 2019. Available online: https://www.cdc.gov/tobacco/data_statistics/fact_sheets/adult_data/cig_smoking/index.htm. Accessed date: 15 January 2021.
10. Ritchie H, Roser M. Smoking. 2013. Published online at OurWorldInData.org. Available at: https://ourworldindata.org/smoking#the-rise-and-fall-of-smoking-in-today-s-rich-countries. Accessed date: 15 January 2021. [Online Resource].
15. Saliba, W., Rennert, H.S., Gronich, N., Gruber, S.B., Rennert, G. Red meat and processed meat intake and risk of colorectal cancer: a population-based case–control study. Eur J Cancer Prev 2019; 28, 287–293. doi:10.1097/cej.0000000000000451
Author Response
Thanks a lot for reviewing our article.
- The abstract paragraph was rephrased and the mathematical formula was omitted (it appears only in the "Methods" section).
- The sentence was corrected accordingly ("Potentially modifiable cancer risk factors were identified using the literature (2, 3).").
In addition, 95% CI were added to the results (tables). - Table 5 and table 6 were transferred to the "Results" section. Now the discussion section contains no tables.
- Indeed, references 1, 4, 5, 7 are in Hebrew and not available in English. However, these references are important, especially for future Israeli researchers that may read this article, and therefore we provided these references, mentioning that they are in Hebrew.
In addition, references 8, 11, 12, 13, 14 were corrected and appropriately written.
Finally, the reference section was revised in accordance with the journal rules.